# Measurement of Human Body Segment Properties Using Low-Cost RGB-D Cameras

**DOI:** 10.3390/s25051515

**Published:** 2025-02-28

**Authors:** Cristina Nuzzi, Marco Ghidelli, Alessandro Luchetti, Matteo Zanetti, Francesco Crenna, Matteo Lancini

**Affiliations:** 1Department of Mechanical and Industrial Engineering (DIMI), University of Brescia, Via Branze 38, 25123 Brescia, Italy; 2Department of Medical and Surgical Specialties, Radiological Sciences, and Public Health (DSMC), University of Brescia, Viale Europa 11, 25123 Brescia, Italy; marco.ghidelli@unibs.it (M.G.); matteo.lancini@unibs.it (M.L.); 3Department of Industrial Engineering (DII), University of Trento, Via Sommarive 9, 38123 Trento, Italy; alessandro.luchetti@unitn.it (A.L.); matteo.zanetti@unitn.it (M.Z.); 4Department of Mechanical Engineering, Energetics, Management and Transportation (DIME), University of Genova, Via alla Opera Pia 15, 16145 Genova, Italy; francesco.crenna@unige.it

**Keywords:** body segment parameters, measurement science, anthropometry, biomechanics, body volume estimation, Kinect Azure

## Abstract

An open question for the biomechanical research community is accurate estimation of the volume and mass of each body segment of the human body, especially when indirect measurements are based on biomechanical modeling. Traditional methods involve the adoption of anthropometric tables, which describe only the average human shape, or manual measurements, which are time-consuming and depend on the operator. We propose a novel method based on the acquisition of a 3D scan of a subject’s body, which is obtained using a consumer-end RGB-D camera. The body segments’ separation is obtained by combining the body skeleton estimation of BlazePose with a biomechanical-coherent skeletal model, which is defined according to the literature. The volume of each body segment is computed using a 3D Monte Carlo procedure. Results were compared with manual measurement by experts, anthropometric tables, and a model leveraging truncated cone approximations, showing good adherence to reference data with minimal differences (ranging from +0.5 to −1.0 dm^3^ for the upper limbs, −0.1 to −4.2 dm^3^ for the thighs, and −0.4 to −2.3 dm^3^ for the shanks). In addition, we propose a novel indicator based on the computation of equivalent diameters for each body segment, highlighting the importance of gender-specific biomechanical models to account for the chest and pelvis areas of female subjects.

## 1. Introduction

The topic of human volume estimation is important for several fields, including healthcare and biomechanics, and it also extends to commercial purposes related to the clothing market and entertainment. In biomechanics, it is common to represent the human body using rigid links to simplify the calculation of kinetic properties through inverse dynamics [1]. However, this process is susceptible to the values of the body segments, especially their mass and center of mass, as they significantly influence the outcome [2]. Obtaining direct measurements of these parameters typically requires medical imaging systems, such as computerized axial tomography scans. Unfortunately, this technology is inaccessible to many. As a result, the research community tried to close the gap by proposing new, affordable methods to achieve comparable results. A cost-effective alternative involves the utilization of consumer-grade 3D cameras like the Kinect, which was explored with success in the biomechanics literature [3,4]. However, these methods typically necessitate a full scan of the subject, encompassing both front and rear surfaces, which can be challenging or impossible for individuals with motor impairments who cannot stand upright [5]. For example, Pfitzner et al. used a Kinect v1 camera to acquire the point cloud of several subjects lying on a bed, standing, and walking [6]. The objective, in this case, was to obtain an estimation of the total weight of the subject through an artificial neural network using, without computing body segments or volumes, the 3D point cloud as input data. Recent endeavors have also proposed alternative 3D imaging-based methods, as outlined in [7]. The work in [8] is the most similar to the proposed one. However, the authors obtained the full 3D scan of subjects that were standing still by rotating a Kinect v2 camera around their bodies. The complete point cloud was then separated into body segments with a similar division as the one that was adopted in our work, but they were manually highlighted using colored pens and tapes on the subjects’ bodies. These landmarks were used by the authors to separate the body segments in the point cloud afterward.

Another viable option is utilizing optimization techniques on specific movements performed by subjects [9]; however, this approach is limited to healthy individuals due to the physical demands of the tests, which can be excessive for those with injuries. Another method for estimating volume, albeit somewhat invasive, is water displacement, which offers greater accuracy [10]. However, measuring each body segment of living subjects with this method is impractical. As a result, researchers often resort to estimating body segment parameters using data from the existing literature [11]. These estimates typically rely on regression equations derived from cadaver studies [12,13] or imaging data [14], along with the subject’s height and weight. Nonetheless, these data sources are population-specific, may lack necessary measurements, especially for specific groups like women or different populations, and can introduce substantial approximations [15,16]. These differences are even more evident when applied to subjects with locomotor impairments, such as spinal cord injury survivors [17]. In certain cases, manual measurements conducted by trained personnel using standard tape measures are employed to create subject-specific models, as suggested in [2]. These measurements are then used to estimate volume by applying a truncated cone model, which approximates the actual segment shape, especially for body regions like the shoulders, trunk, and pelvis [18]. However, this approach has limitations, including reliance on operator accuracy and precision, as well as being time consuming [19].

Other modern approaches extensively leverage deep learning models to obtain an estimation of the human volume. Focusing on the entertainment and clothing markets, the authors in [20] proposed a deep learning method to generate body segment measurements (e.g., the torso circumference) from a set of input data, including both 2D and 3D whole-body images. Another similar work is [21], where the authors used a single stereo camera to acquire the depth map of a subject lying on a bed. However, that work only focused on the computation of the overall volume and height of the scanned subjects by analyzing the convex hull of the sub-portions of the overall point cloud. Similarly, Hu et al. developed a novel method based on deep learning to estimate a scanned person’s whole 3D point cloud, which separates the body segments automatically [22]. However, only a few body segments were chosen (arms, legs, torso, and head), and the method focuses on people standing still. Although advanced, methods based on deep learning techniques are often trained on synthetic 3D models and tested on public 3D datasets that usually do not include subject-specific volume, mass, and segment length information. Consequently, the performances of such methods were not evaluated in comparison with healthcare practices, such as anthropometric tables or manual measurements, and they are more suited for the commercial world than the healthcare and biomechanics ones.

To address these challenges, we propose an innovative and low-cost method for estimating the volume and mass of a subject’s body segments. This method involves capturing and processing 3D point clouds and color data, leveraging a deep learning skeletonization model. The proposed work focuses on people who cannot stand; thus, they cannot be subjected to time-consuming acquisition procedures or measurement methods that are as sophisticated as they are expensive and, therefore, not widely used. The measurement setup design is intended to be used for this kind of patient and, at the same time, offers a low-cost and rapid measurement of their body segments’ volume and mass. The measurement conditions will necessarily have to be adapted to the needs of the patients, and, considering the pathologies we are addressing, the only viable option is to measure the patients while they were lying on a comfortable horizontal surface, which is readily available in a hospital setting. In particular, in tasks such as biomechanical analyses of this kind of patient during motion (e.g., inverse kinematics), the anthropometric characteristics of the limbs cannot be assimilated to those of healthy people, so they cannot be obtained from well-known anthropometric tables such as [12]. To advance this specific research field, it is therefore important to obtain a reliable measure of a limb’s volume and mass to produce good results through inverse kinematics [23]. However, before conducting extensive experiments on patients, it is of utmost importance to first validate the measurement pipeline proposed on healthy subjects. This is the reason why the present work describes the validation of the developed measurement pipeline on healthy subjects, focusing on the comparison with existing methods used by healthcare professionals and by the biomechanics research community.

Another key aspect of the proposed work concerns the biomechanical model and the definitions of the body segments. Typically, in the field of biomechanics, the human body segments adopted are obtained through an approximation that simplifies the consequent calculations, rather than realistic models of a human body such as those obtained through medical tomography. As a consequence, in this work, the biomechanical model adopted is the standard model documented in the EUROBENCH project [24].

The initial version of this study was introduced in [25]. In this extended version, the main advancements proposed in contrast with the existing literature are as follows:1.The acquisition setup adopts a single Kinect Azure, which is an RGB-D camera with enhanced performance compared with its predecessors, positioned atop the experimental bed, thus producing a refined subject point cloud with reduced noise [26].2.A total of 60 healthy subjects (30 males and 30 females) participated in this study (two times more than the previous work). It is worth noting that conducting measurement tests on a gender-balanced dataset is uncommon in the research community.3.A biomechanical model based on the combination of the skeletonization model BlazePose [27] and the biomechanical segments separation model defined by EUROBENCH project [24] is proposed. Its robustness is evaluated in a dedicated experiment for both male and female subjects.4.The proposed volume estimation method is based on a Monte Carlo generation of points inside the alpha shape approximation of a 3D body segment, a procedure that any computer can conduct without employing heavy and complex deep learning models. Consequently, the output volume is the result of a measurement conducted on the subject body, in contrast with generalized anthropometric tables.5.The resulting gender-specific data are validated by comparing them with the anthropometric tables [12] and manual measurements, especially focusing on the limbs since they account for the majority of the inertia during kinematic analysis of moving subjects. In addition to the volume comparison, we propose a novel method based on the computation of “equivalent diameters” to better analyze the model’s results. The evaluation part of this article is the strong feature of our study, whihc is often overlooked in the majority of articles on the matter.

The acquired point cloud and color frames undergo a post-processing procedure that quickly provides estimates of body segment volumes and masses for each subject. A scheme of the proposed analysis is shown in Figure 1. The proposed approach can be of use for several works related to the human body volume and mass estimation, e.g., measuring the thorax expansion during respiratory phases [28]; obtaining personalized anatomy of subjects for medical purposes [29] and virtual reality [30]; body segment impedance analysis, which is a typical data point used in the field of biomechanics [31]; and aid in the estimation of the dynamics of exoskeleton users during movement [32,33,34].

## 2. Materials and Methods

### 2.1. Experimental Protocol

Before starting the experiment, each subject was informed about the research aim, the experiment and protocol procedure, and their rights according to the Declaration of Helsinki. The subjects were then asked to undress, keeping only their underwear (and the brassiere for women), removing unnecessary devices, such as watches, necklaces, bracelets, glasses, and earrings, and wearing an elastic cap to prevent hair from being identified as a body segment. A trained staff member measured the subject’s body using a sartorial tape. This procedure was conducted in compliance with standard healthcare practices that require the subject to stand to allow for the identification of its skeleton joints. It is worth noting that, as reported by the literature [35,36,37], there is minimal difference between taking a manual measurement while the subject is standing versus while lying horizontally, especially for subjects with low adipose tissue, which was the case in our experiment. The subject’s manual measurements used as reference measurements are summarized in Table 1, and they are also reported in [25]. Measurements were always taken considering the front side of the subject’s body, except for those cases where it was stated otherwise in the description. Three types of measurements were taken:1.Length measurements: these were taken considering the body segment’s length (*L*) from the distal and proximal joints using sartorial tape.2.Width measurements: these were taken using the anthropometric compass to measure the distances (*D*) between joints or the width (*W*) of the body segment from side to side (usually from the front) corresponding to specific joint positions.3.Circumferences: these were taken using sartorial tape in correspondence with specific joints or when the body segment is circular (*O*).

After the manual measurements were taken, the subject was required to lie on the bed for the 3D data acquisition, keeping the same posture shown in Figure 2b. During the volume acquisition procedure, the subject was asked to exhale all the air from the lungs to avoid detecting an over-expanded chest, as suggested in [38]. The operator counted backward to inform the subject of the start of the acquisition. During acquisition, the subject held their breath until the end of the acquisition process, which took about three seconds.

### 2.2. Experimental Campaign

The experimental campaign started in December 2022 and ended in May 2024 at the University of Brescia. A total of 60 healthy participants (50% males and 50% females) were recruited. The inclusion criteria were an age of 18–79 years and the ability to give informed consent. The exclusion criteria were an inability to lay down and stay still on the bed, uneasiness in undressing down to underwear, and any condition affecting the body shape (e.g., amputations and pregnancy). Participants were informed about the experimental procedures and gave their written consent in accordance with the principles of the Declaration of Helsinki. No personal data were stored except for those needed for the tests. Table 2 shows the averages and standard deviations of the age, height, mass, and body mass index (BMI) of the subjects, according to gender.

### 2.3. Materials

The system used for the acquisition was the experimental orthopedic bed used in [25]. The size of the T-shaped bed was 106×206×39 cm, and it was equipped with a wooden frame to avoid lateral movements of the structure when the subjects lay supine on the bed. To ensure the subjects’ comfort, a custom-made, low-deformation foam mattress of a density equal to 0.1 kg/dm^3^ was placed on top of the wooden frame. It is worth noting that the adoption of the orthopedic bed as the measurement position instead of a vertical upright position was motivated by the target application, which involves patients with motion impairments, i.e., those unable to stand still.

A Kinect Azure RGB-D camera [39] was mounted above the center of the bed using a steel frame, ensuring the entire mattress was inside the camera’s field of view. The camera was equipped with (i) a color sensor with a resolution equal to 1920×1080 px, an aspect ratio of 16:9, and an angular field of view of 90×59 degrees; and (ii) a depth sensor based on time-of-flight technology, operated in NFOV unbinned mode, resulting in a depth map resolution of 640×576 px that corresponds to an angular field of view of 75×65 degrees in the operating range of 0.5–3.86 m. The camera was calibrated and warmed up following the procedure detailed in [26].

The camera was connected to a PC to collect, save, and process the subjects’ data, which were acquired using Kinect Azure’s SDK. For each test, a video lasting about 2 s was recorded using the MKV format, which contains the color and depth frames and the camera parameters useful for post-processing, such as the camera’s calibration matrix, timestamps, and acquisition mode. It is worth noting that, since the MKV of the acquisition contains a sequence of frames, the consistency of the images used can be verified (e.g., the subject and the camera do not move, data are valid, and no reflections or occlusions appear). For each recording, the first valid pair of color and depth frames were extracted and converted into a point cloud.

### 2.4. Body Skeleton Estimation

The subject’s skeleton was estimated using MediaPipe BlazePose [27], which is considered a state-of-the-art technology among pose estimation models [40] thanks to its speed, computational requirements, and detection accuracy. BlazePose outputs a total of 33 keypoints (KPs); however, these KPs do not correspond to the actual body joints of the human skeleton. This is the reason why we selected only a portion of BlazePose’s KPs (13 in total, shown in red in Figure 2a), which will then be used to compute new body joints that are compliant with the biomechanical definition of the human skeleton in the post-processing step, as detailed in Section 2.5 [24]. The color frame (resolution of 1920×1080 px) of the subject extracted from its corresponding MKV file was elaborated by BlazePose, and the 13 KPs of the subject’s body were saved in a JSON file. From the original 13 BlazePose KPs, two more were computed (as shown in yellow in Figure 2a): the neck joint KP2 (the midpoint of the shoulders) and KP15 (the midpoint of the hips).

### 2.5. Biomechanical Model

Pose estimators like BlazePose do not estimate the actual human skeleton, producing instead a rough estimate of the position of fundamental body joints. Therefore, the body segment (BS) definition derived from the BlazePose skeleton was corrected using the biomechanical model definition used in the EUROBENCH project [24]. This model is based on the anthropometric data reported in [12] and on [41], which describe the BS dimensions and the average vertebrae heights, respectively. Hence, the resulting corrected model applied in this work, and shown in Figure 2a, presents two more KPs (KP16 to KP19, in black), which are obtained as proportions of the average vertebrae heights (as in [41]) and on the BS dimensions (as in [12]). The procedure is as follows:1.From the original BlazePose KPs (the red KPs in Figure 2a), the neck joint KP2 is computed as the midpoint of the shoulders (KP3 and KP6), and KP15 is computed as the midpoint of the hips (KP9 and KP12). They are depicted in yellow in Figure 2a.2.A vector midBody is computed by joining KP2 and KP15.3.The position of C7 (KP19) is estimated starting from the neck joint (KP2) and moving upward a fraction of midBody, which is estimated according to the angle between the suprasternal notch and the cervical joint center, as reported in [13]:(1)$$KP19=KP2+\frac{KP1-KP2}{\left|KP1-KP2\right|}\cdot \left|midBody\right|\cdot 17\%.$$4.KP17 represents the midpoint between the anterior superior iliac spines, and it is computed using the normative proportion of the hip that was reported in [12] by moving KP15 (mid hip) upward by a fraction of the width of the hip $W_{hip}$:(2)$$KP17=KP15+\frac{midBody}{\left|midBody\right|}\cdot W_{hip}\cdot 47\%.$$5.The location of the last thoracic vertebra, $T_{12}$ (KP18), is computed as a point on the midBody at a fraction of its length, as determined in [12]:(3)KP18=KP17+midBody·36%.

The resulting 19 KPs are expressed in pixel coordinates, referring to the 2D camera frame centered on the upper-left corner of the color image.

### 2.6. KP Coordinate Conversion

The main reason for using RGB-D cameras in an application is that they allow fo the acquisition of color and depth frames that are temporally and spatially aligned. The procedure that allows for the re-projecting of 2D points in 3D space and vice versa, from which the colored point cloud is computed, is called image registration [42]. Despite being a well-known principle, in practice, careful attention should be paid to the transformations needed to undistort both the color and depth frames. In fact, according to the camera, the lenses adopted by the two sensors of the device may be different, thus introducing lens distortions in the two frames that may interfere with the proper overlapping of the two fields of view. Lens distortions should be individually estimated and corrected for each sensor before applying the rigid transformation procedure that matches the two frames by rotating, stretching, and translating them through a transformation matrix *T* so that they can perfectly overlap. Unfortunately, the process of undistorting and matching the two frames can differ according to the camera manufacturer, so it may not be straightforward. For example, the Kinect v2 used in [25] heavily relies on the camera’s lookup tables to properly compute the depth frame’s distortion parameters. However, this procedure is coded in the source code of the camera’s SDK. As a result, the research community developed workarounds to account for lens distortions faster and without the hassle of using the proprietary code. This was the case in our previous work, in which we adopted an approximated formula to convert 2D pixel coordinates to corresponding 3D coordinates. Being an approximation, using this approach was a potential source of error that we aimed to correct in this upgraded version. In contrast, Azure Kinect SDK and the Python wrapper pyk4a [43] allow users to directly apply the proprietary registration procedure through using specific internal functions that convert 2D coordinates into 3D space by leveraging the camera’s internal parameters saved in the acquisition video, which are saved in MKV format. This is especially important for Kinect Azure since the lens distortion of the depth sensor is quite severe. Therefore, after we obtained the 19 KPs of the biomechanical model in pixel coordinates, we loaded them in a separate Python function that opens the subject’s acquisition MKV, undistorts the frames using the two sensors’ fundamental matrices, transforms the depth frame to match the color frame, and finally applies the coordinate conversion to the points. At the end of this procedure, we obtained a file containing the 19 KPs expressed in metric coordinates, which were saved in JSON format. It is worth noting that the KPs obtained were already in the correct position over the subjects’ bodies since they were expressed in the same reference system as the subjects’ point clouds; thus, no transformation was required. This was possible thanks to the internal calibration of the camera conducted beforehand, which allowed the depth and color sensors to be spatially aligned.

### 2.7. Point Cloud Filtering

Due to the camera’s field of view, the acquired point cloud ($PC_{raw}$, Figure 3a) inevitably included elements of the bed and the background, which interfered with a proper volume estimation procedure. Therefore, $PC_{raw}$ undergoes a filtering procedure composed of two steps: coarse filtering, which removes unwanted data from $PC_{raw}$; and fine filtering, which aligns the result to the reference system axes. The resulting point cloud is a body figure without the bed’s and subject’s back points, which is left empty because it lies on the bed’s surface. As a result, the point cloud is not a closed object.

#### 2.7.1. Coarse Filtering

This step allows for the removal of unwanted points of $PC_{raw}$ that fall outside the bed’s area. To do so, we first generated an area of interest in the XY plane containing only the bed and the subject’s point cloud. The area is a rectangle whose dimension and position are automatically computed by the following procedure:1.Calculate the minimum and maximum values $min_{X}$, $min_{Y}$, $max_{X}$, and $max_{Y}$ along both X and Y coordinates.2.Correct them by adding or subtracting the padding of known values to properly discard points belonging to the mechanical structure and keep those belonging to the bed’s area.After removing the points outside the XY area of interest, we computed the mean Z value of the point cloud, to which we add a padding of 40 cm to properly delete the points belonging to the ground and the eventual noise around the bed’s area. Points with a Z coordinate higher than this value were discarded. It is worth noting that the reference system was centered in the camera frame; hence, the zero position was around the center of the bed (in the XY plane) and at the height of around 2 m, which is the Z distance of the camera from the ground. The result was $PC_{coarse,0}$, as shown in Figure 3b.

#### 2.7.2. Fine Filtering

Although the camera was fixed on the bed’s mechanical structure, it was not easy to ensure a perfect Z-axis alignment. Moreover, when a subject lies on the bed, it is also inevitable that the reference system will be slightly misaligned along the X and Y axes. The procedure is as follows:1.Apply a plane fitting on $PC_{coarse,0}$ to extract the bed’s plane normal, which is aligned to the upward direction of the Z reference axis. This results in a transformation matrix $T_{1}$ and a transformed point cloud $PC_{coarse,1}$ (Figure 4a).2.From the bed’s point cloud of $PC_{coarse,1}$, create a 2D rectangular region that inscribes the bed using its minimum and maximum values along X and Y. The rectangular region is then filled with $10^{6}$ Monte Carlo points.3.Conduct a principal component analysis (PCA) on the rectangular region that approximates the bed, thus finding its orientation along the X and Y axes.4.Align the rectangular region principal components to the reference axes X and Y. This results in the transformation matrix $T_{2}$ that, when applied to $PC_{coarse,1}$, outputs $PC_{coarse,2}$ (Figure 4b).5.Perform another plane fitting on $PC_{coarse,2}$ to find and remove the points belonging to the bed’s plane, thus obtaining only the points of the subject’s body.6.The resulting point cloud is then filtered by applying an outlier removal procedure based on RANSAC, which is followed by a denoising step. The final point cloud obtained is $PC_{fine}$ (Figure 3c).

The output of the whole procedure is the transformation matrix $T=T_{1}\cdot T_{2}$ and the filtered point cloud $PC_{fine}$. Thanks to the applied filtering steps, the point cloud orientation is corrected and aligned to the XY plane to facilitate the projection of its points onto it. For the same reason, the 19 KPs in 3D coordinates obtained at the end of the conversion step, as detailed in Section 2.6, were transformed to the reference system $RF_{corrected}$, and the transformation matrix *T* was also used for this process. These were the two necessary steps for the body segments’ (BSs’) separation procedure, as detailed in Section 2.8.

### 2.8. Body Segment Separation

After the alignment corrections, as described in Section 2.7, the resulting $PC_{fine}$ was centered in the new reference frame $RF_{corrected}$, and it was possible to project the points onto the 2D XY plane without losing data or adding deformations. As a result, to separate the body parts, it is easy to define a 2D geometrical shape that includes all the BS points. These shapes are called enclosing polygons (EPs). Thanks to the vectors and the KPs obtained from the biomechanical model (Section 2.5, Figure 2b), each EP can be built considering its proximal and distal KP from which the corresponding vectors generate. Then, the complete EP is obtained by mathematically finding the intersection between these vectors and eventually generating the parallel and/or orthogonal lines passing through the KPs according to the EP shape. For most BSs, the EP is determined by the intersection of 4 to 5 lines using simple mathematical equations (e.g., arms, forearms, legs, shanks, abdomen, and thorax). The other segments require an ad hoc routine that cannot be easily generalized; e.g., the shoulders are determined by the intersection of 8 lines, while the head, the hands, and the feet are determined considering their proximal KPs as the starting point and the vectors generated from it. Since they do not possess a distal KP to close the polygon, it is created as a copy of the proximal KP moved at a fixed empirical offset, which is different for each BS. Examples of the generated EPs are shown in Figure 5, while the vectors used to generate the EPs from KPs are visible in Figure 2. Figure 6a shows an example of the result where the KPs are drawn on top of $PC_{fine}$.

### 2.9. Body Segment Point Cloud Filling

Since the bottom of the resulting $PC_{fine}$ is empty due to the removal of the bed’s surface (Section 2.7), for each BS’ point cloud, some artificial points are generated by copying the original points of the BS’ point cloud and projecting them to the maximum Z coordinate that coordinates with the position of the removed bed. This step is named bottom filling. However, some BSs may have wide gaps that interfere with the volume computation (e.g., the gap between the arm and the forearm that corresponds to the elbow KP). When the gaps are too wide or the shape of the body segment’s point cloud has a pointy geometry (e.g., shoulders and trunk), artificial points at different Z coordinates are created starting from the XY coordinates of points that belong to the surface’s edge. This step is called gaps filling. An example of the resulting point cloud of this process is shown in Figure 7a.

### 2.10. Volume and Mass Computation

To estimate the volume of a BS using the Monte Carlo procedure, as detailed in [44], the idea is to enclose the BS’ point cloud in a rectangular cuboid. Each size of the cuboid and the coordinates of its corners are computed considering the minimum and maximum dimensions of the BS’ point cloud. The cuboid is then filled with $N\geq 10^{6}$ Monte Carlo points. The BS’ volume is calculated as follows, and it is expressed in m^3^:(4)$$V_{mis,BS}=\frac{N_{BS}}{N}\cdot V_{cuboid},$$
where $V_{cuboid}$ is the volume of the cuboid, and $N_{BS}$ corresponds to the number of Monte Carlo points that fall inside the BS’ point cloud. To reduce the computation time compared to the algorithm in [44], our approach leverages alpha shapes to estimate $N_{BS}$. The alpha shape of a set of points representing an object is a description of the corresponding shape, either in 2D or 3D space. It is a generalization of the convex hull and a subgraph of the Delaunay triangulation. The alpha parameter is defined as the value *a*, such that an edge of a disk of radius 1/a can be drawn between any two edge members of a set of points and still contain all of the points [45]. It is worth noting that to correctly compute the alpha shape of the BS, the point cloud must not have large gaps (see Section 2.9). An example of its appearance is shown in Figure 7b. For each BS, the *a* parameter values used in this work were found experimentally and are as follows: (1) 0.08 for the head, shoulders, pelvis, arms, shanks, and feet, (2) 0.05 for the forearms and hands, (3) 0.15 for the trunk, (4) 0.2 for the abdomen, and (5) 0.22 for the legs.

The value of $N_{BS}$ is obtained by converting the alpha shape to a polyhedron and counting which Monte Carlo points are inside it (see Figure 7c for an example). Finally, to compute the mass of each segment (in kg), it is sufficient to multiply the volume by a density value as follows:(5)$$M_{mis,BS}=\rho_{ref,BS}\cdot V_{mis,BS}.$$

In this work, the BSs’ density $\rho_{ref,BS}$ was taken from the literature [12] if available; otherwise, a value of 1000 kg/m^3^ was considered instead.

## 3. Results and Discussion

### 3.1. Biomechanical Model Robustness

The accuracy of the KPs’ estimations obtained from Mediapipe BlazePose was ensured by the corresponding literature [27,40], demonstrating that the KPs’ positioning in color images closely followed the manual labeling that was used as the ground truth. However, we modified the resulting BlazePose skeleton to obtain our biomechanical model (as described in Section 2.5). To verify the robustness of this approach, we compared the results of the BSs’ separation, which was performed on the 3D image, with the body BSs’ separation, which was performed manually by an expert drawing section lines on the skin of the subjects. We applied the proposed procedure two times to two subjects (1 male and 1 female): both with and without section lines drawn (the marked images are shown in Figure 8 for both subjects). The manual markings were applied to their bodies by an expert therapist after the unmarked data were obtained using a marker pen for the female subject and a tape for the male subject. Between the two acquisitions, the subjects were asked to lay still to reduce possible variability (e.g., positioning of the body). In either case, the positioning of the KPs was not altered by the presence of manual markings. Overall, the model closely followed the markings, with a difference of only a few centimeters between the real position of KP17 and KP18 and the position obtained by our model (−1.6 cm for KP17 and +1.3 cm for KP18 on average), thus confirming similar results that were obtained with multiple images [46]. It is worth noting that, in the case of female subjects, the model did not perform well around the chest area, which produced an error in the separation of the shoulders from the trunk. This was another reason why, in our final analysis, we considered the thorax to be equal to the sum of the shoulders and chest areas instead of keeping them separate.

### 3.2. Output Volume

The proposed measurement approach outputs the volume and mass of each subject’s BS, $V_{scanner,BS}$. To ensure that these results are comparable with the literature (the anthropometric tables in [12]) and the manual measurements of the subject, we compared the limb (arms, forearms, thighs, and shanks) volumes obtained by our scanner method with the literature values ($V_{ref,BS}$) and with the volume of a truncated cone that approximates the BS ($V_{cone,BS}$). To do so, we computed the ΔV values as follows:(6)$$\begin{array}{cc} \Delta V_{scan-ref,BS} & =V_{scanner,BS}-V_{ref,BS} \end{array}$$(7)$$\begin{array}{cc} \Delta V_{scan-cone,BS} & =V_{scanner,BS}-V_{cone,BS}. \end{array}$$

The volume of a BS (in dm^3^) from the reference tables was obtained as follows:(8)$$V_{ref,BS}=M_{ref,BS}\cdot \rho_{ref,BS},$$
where $M_{ref,BS}$ is the BS mass, and $\rho_{ref,BS}$ is the BS density, both as reported in [12]. The volume of the BS obtained by approximating the BS shape to a truncated cone is as follows:(9)$$V_{cone,BS}=\frac{1}{3}\cdot L_{manual,BS}\cdot \pi \cdot \left(r_{d}^{2}+r_{p}^{2}+r_{p}*r_{d}\right),$$
where $L_{manual,BS}$ is the BS length, as manually measured, and $r_{d}$ and $r_{p}$ are the radii of the distal and proximal circumferences of the BS, respectively.

Figure 9a shows the difference between the limbs’ volumes obtained by our scanner approach and the reference tables ($\Delta V_{scanner-ref,BS}$), while Figure 9b shows the difference between the limbs’ volume obtained by our scanner approach and the truncated cone approximation ($\Delta V_{scanner-cone,BS}$). Values were obtained for each subject and visualized according to gender (blue boxes refer to male subjects and pink boxes to female subjects). The results highlight that, in both cases, the difference for the upper limbs is around 0 dm^3^ for both males and females. For the thigh volume, the proposed scanner method underestimates the females’ thigh volume more than the males’ (the difference spans from −0.1 to −2.5 dm^3^ for males and from −1.5 to −4.2 dm^3^ for females with respect to Figure 9a, and +0.4 to −3.0 dm^3^ for males and −0.5 tp −5.0 dm^3^ for females with respect to Figure 9b). This underestimation may be due to two reasons: (i) the back portion of the leg’s point cloud was missing, thus it may have beeen highly approximated, especially for less muscular subjects; and (ii) the pelvis area was not always selected correctly for the female subjects because a portion of the upper part of the legs was considered part of the pelvis instead of the thigh. This is also evident looking at Figure 8.

### 3.3. Comparison of the Body Segment Lengths

We compared the BS lengths obtained by BlazePose (which was modified by our biomechanical model) with the reference tables and the manual measurements, respectively, as shown in Figure 10a,b, and these were then normalized over the subject’s height (SH). Normalization was necessary to understand and compare the results because the same quantity had a different impact according to the subject’s physique. We excluded the head, the pelvis, the hands, and the feet BSs because the KPs of these segments did not coincide with their endpoints (see Figure 2). This resulted in ΔL values named $\Delta L_{scanner-ref,BS}$ and $\Delta L_{scanner-cone,BS}$, respectively.

The BS lengths computed by our model are close to the ones reported in [12] and to the manual measurements. For the thorax BS, the length difference was underestimated, especially when compared to the manual measurements. This issue was probably due to the manual measurements of the upper portion of the back (C7-T10 for the thorax and T10-L5 for the abdomen), which were taken considering the spine’s curvature; however, when the subjects lay on the bed, the KPs were estimated on the front, thus resulting in an estimation error. Moreover, since the manual measurements were taken when the subject was standing, the body shape was modified due to adipose tissue relaxation when lying on the bed, and, thus, the two volumes differed. However, this issue is characteristic of the measurement setup design, which is aimed at subjects with motion impairments; thus, the bed is a necessary equipment. Despite this issue, it is worth noting that the main difference for $\Delta L_{scanner-ref,thorax}$ was in the results of the female subjects, probably due to the presence of the brasserie that produced skeleton estimation issues, as discussed in Section 3.1. As for the shanks, the underestimation may be due to the position of the legs during the experiment, which were slightly open and with the knees pointed laterally instead of up for several of the subjects. For the female thigh, the result may be explained by a wrong positioning of the asi KPs (KP9 and KP12).

### 3.4. Analysis of the Equivalent Diameters

It is worth noting that although the truncated cone approximation could be used in practice to estimate the BS volume of most BSs, it cannot be applied to the head, trunk, abdomen, and pelvis BSs due to their particular shapes, which are not a circular truncated cone. For this reason, we decided to only compare our scanner-based mass estimation with the anthropometric tables [12]. Since the anthropometric tables represent only the average body distribution and do not consider the specific body shape of each subject, an *equivalent diameter* was computed for each BS as a proxy of its actual shape. This metric corresponds, for each BS, to the diameter of the cylinder generated as the approximating geometric solid with (i) the same mass and density estimated using the anthropometric tables, and (ii) the actual length of the BS measured by hand. Hence, the equivalent diameter (in mm) of the BS was obtained as follows:(10)$$D_{ref,BS}=\sqrt{\frac{4}{\pi }\cdot \frac{V_{ref,BS}}{L_{manual,BS}}},$$
where $L_{manual,BS}$ is the BS’s length that is manually measured by an expert operator. Similarly, the equivalent diameter of a BS from our scanner is as follows:(11)$$D_{scan,BS}=\sqrt{\frac{4}{\pi }\cdot \frac{V_{scan,BS}}{L_{scan,BS}}},$$
where $V_{scan,BS}$ and $L_{scan,BS}$ are the volume and length of the BS obtained from the scanner, respectively. The normalized diameters ${\hat{D}}_{ref,BS}$ and ${\hat{D}}_{scan,BS}$ are, therefore, obtained by normalizing both of the values for the corresponding $L_{manual,BS}$. Therefore, the difference between the two diameters, which is normalized over the subject’s height, can be computed as follows:(12)$$\Delta D_{BS}=\frac{\left({\hat{D}}_{scan,BS}-{\hat{D}}_{ref,BS}\right)}{H_{subject}}.$$The resulting $\Delta D_{BS}$ is shown in Figure 11a for the thorax, abdomen, arm, forearm, thigh, and shank BSs. The thorax and abdomen BSs’ behavior is explained by the difference in the scanned volume (due to the absence of the back’s points) and BS length.

In Figure 12a, the distribution of normalized values ${\hat{D}}_{ref,BS}$ and ${\hat{D}}_{scan,BS}$ is shown for the thigh BS. Once again, it is evident that the two distributions were comparable with minimal differences, with a slightly increased variability in the diameters computed by the scanner. In addition, Figure 12b shows the Bland–Altman plot of the same BS, which highlights a good balance of the distribution. The results of the other BSs were similar and were thus omitted in this paper for brevity.

For the same BSs, we compared the resulting masses, as shown in Figure 11b, and these were normalized over the subjects’ body weight (BW). The data highlight an underestimation of the thorax’s weight that spans from −4.8% to −10.5% for male subjects and from 4.8% to −4.8% for female subjects. A similar error was shown for the abdomen BSs, which were overestimated instead. Once again, this behavior may be due to the absence of the subjects’ back data from the point cloud since the subjects lay on the experimental bed and because of the estimation error of the lengths due to the spine’s curvature. As a result, scanning the whole subject body could solve this issue. However, we want to stress that our application is designed specifically for subjects with motion impairments who cannot stand still, so scanning their whole bodies is impossible. Moreover, in most biomechanics literature, the focus is on the kinematics of the body during movement; hence, the limbs are the BSs with the greatest importance compared to the rest of the body.

## 4. Conclusions

Estimating the mass and volume of the body segments (BSs) of human subjects is an important first step in biomechanical simulations and indirect measurements. The research community typically estimates them using anthropometric tables that are often unable to represent the specific population involved in the tests or a specific subject with an uncommon body type (SCI, amputees, athletes, etc.), or by acquiring manual measurements of a subject’s BSs using sartorial tape. This second approach is the most commonly used in practice; however, it is affected by operator error [19]. Therefore, we propose a novel approach for measuring a subject’s volume by obtaining a 3D point cloud from an affordable RGB-D camera, Kinect Azure. The proposed measurement system is specifically designed to work with subjects with motion impairments. The present work describes the experimental campaign and results conducted to validate the system on healthy subjects: a fundamental requirement before conducting extensive research on the target patients.

The method separates the BSs by applying a ready-for-use skeletal model that is computed starting from the state-of-the-art skeletonization model MediaPipe BlazePose. The proposed biomechanical model’s robustness was ensured by checking its accuracy in the positioning of the KPs on the color images compared to the manual markings made by an expert. The resulting volumes and lengths of the BSs were compared with those obtained from reference tables [12] and by approximating the BSs to truncated cones. This shows that our approach is robust, with minimal differences between both methods. However, since the truncated cones method does not apply to some BSs (e.g., thorax, abdomen, pelvis, and head), we only compared the mass of the BSs resulting from our method to the corresponding mass reported in anthropometric tables.

A major limitation of this measurement solution is the difficulty in obtaining a reference BS mass information from live subjects. As a workaround, to investigate the accuracy of the mass estimation with subject-specific data, we performed a comparison with the body segments’ equivalent diameters, which were estimated using both the table and manual measurements. Such a comparison showed small differences between the equivalent diameters computed from the anthropometric tables and the ones obtained from the scanner’s data for the limbs. For the thorax and abdomen BSs, they showed higher variability and differences between the male and female subjects due to the presence of the chest and the curvature of the spine, as well as the absence of the back’s points since the subjects lay on the orthopedic bed. These issues also reflect the differences in the mass estimation for the thorax and abdomen BSs, which had the highest variability (from −4.8% to −10.5% for male subjects and from 4.8% to −4.8% for female subjects). These issues could be resolved by acquiring a complete 3D point cloud of the subject [47]; however, the adoption of the orthopedic bed is a necessary requirement for our intended application as it will deal with patients with motion impairments in future works. Moreover, a wide variety of research works in biomechanics focus on the kinematic analysis of such subjects; hence, the BSs with more importance are the limbs instead of the thorax, abdomen, and pelvis BSs. Still, as a future development, we aim to investigate this result further by obtaining a full 3D scan of healthy subjects standing still. Hence, the biomechanical model should be adapted to analyze both the front and back views to mitigate the effect of the spine’s curvature, which is probably the reason why there’s a noticeable difference between the real thorax dimension and the one obtained from the scanner. In addition, our experiments highlighted the necessity of developing gender-specific models to account for the chest and pelvis areas of female subjects [48].

The results highlight that, despite the necessary improvements required to optimize the volume and mass estimation for the trunk, abdomen, and pelvis BSs, the developed system is a valid alternative to traditional practices, such as anthropometric tables and the truncated cones model, even for healthy subjects. As a result, further experiments will involve patients with motion impairments to demonstrate the validity of the developed system on the target subjects. In the case of such subjects, obtaining a full body scan may be difficult or even impossible according to their physical conditions; hence, a workaround to obtain it may be to design dedicated equipment to support them in an upright or partially standing position, coupled with the adoption of markers to facilitate the front and back point cloud fusion.

Finally, in the future, we aim to expand the comparison with truncated cone volumes by defining the method to obtain the corresponding volume of the thorax, abdomen, pelvis, and head BSs, which were excluded from our analysis.

## Figures and Tables

**Figure 1 sensors-25-01515-f001:**
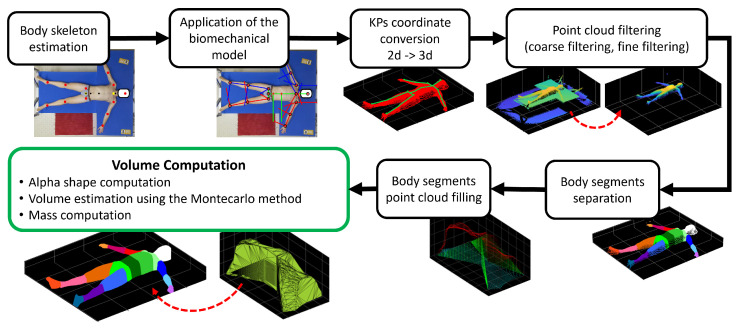
Scheme of the complete processing procedure with example results taken from one subject’s data.

**Figure 2 sensors-25-01515-f002:**
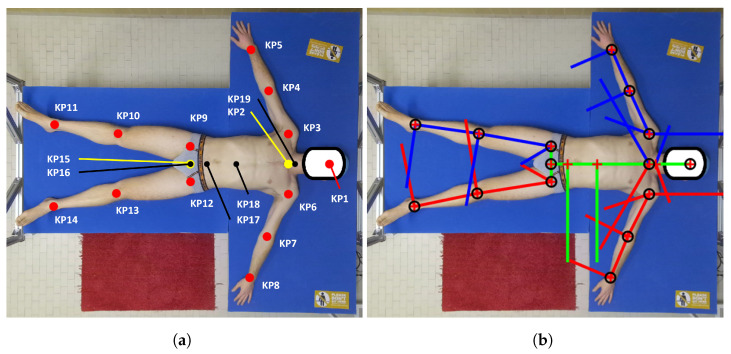
(**a**) Scheme of the KPs considered for this work. Red KPs: computed by BlazePose; yellow KPs: calculated as midpoints of shoulders and hips segments; and black KPs: obtained by applying the biomechanical model to the original BlazePose KPs. (**b**) An example of vectors that were computed by the biomechanical model.

**Figure 3 sensors-25-01515-f003:**
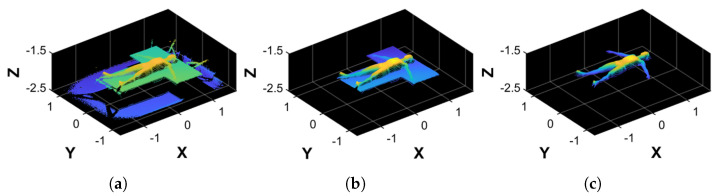
Examples of the filtering process. (**a**) Original point cloud, $PC_{raw}$. (**b**) Result of the coarse filtering process, $PC_{coarse,0}$. (**c**) Result of the fine filtering process, $PC_{fine}$.

**Figure 4 sensors-25-01515-f004:**
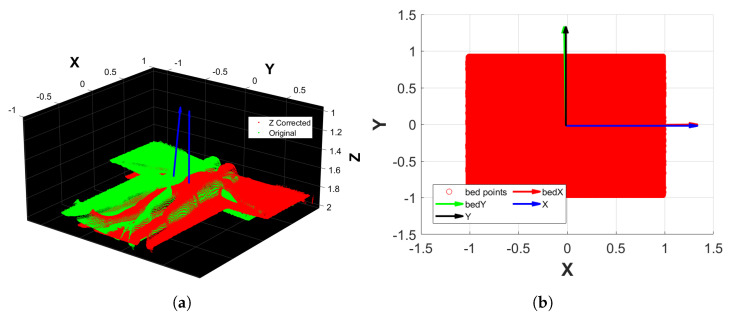
Examples of the orientation correction of the fine filtering process. (**a**) Alignment of the bed’s normal to the Z reference axis. (**b**) Alignment of the bed’s Monte Carlo approximation principal components to the X and Y reference axes.

**Figure 5 sensors-25-01515-f005:**
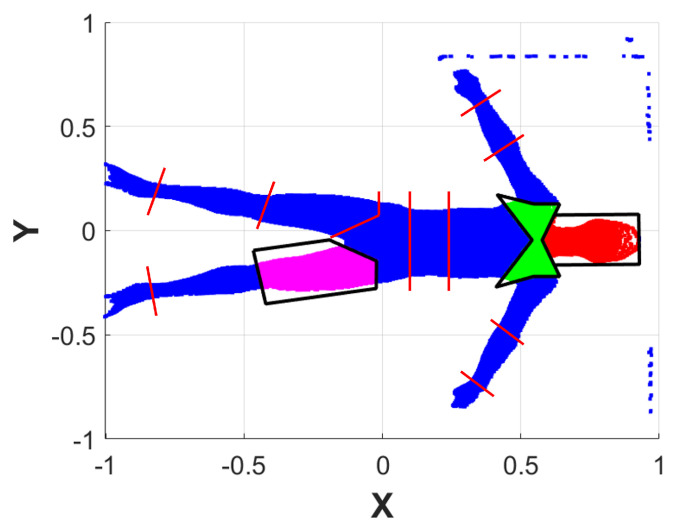
Example of some EPs drawn on a subject’s point cloud projected onto a 2D XY plane. Each EP is shown in black, and the points belonging to them are highlighted in different colors. Vectors used to create the remaining EPs are shown in red.

**Figure 6 sensors-25-01515-f006:**
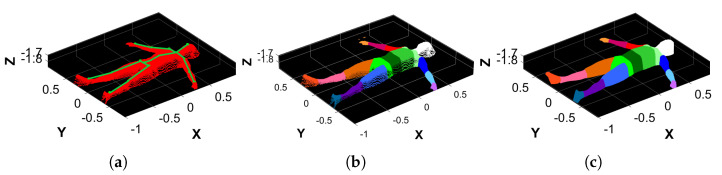
Examples of the body segment separation procedure. (**a**) $PC_{fine}$ of a subject on top, of which the 19 KPs are drawn in green. (**b**) Separation of the BSs of the subject highlighted in different colors. (**c**) Monte Carlo approximation of the BSs, which is necessary to estimate their volume.

**Figure 7 sensors-25-01515-f007:**
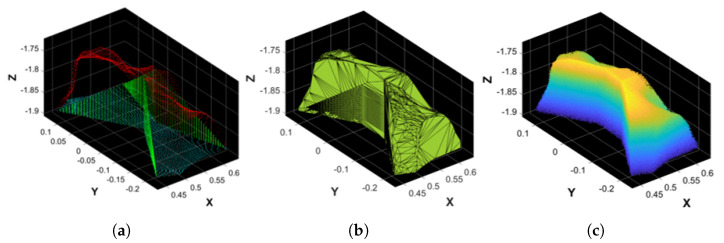
(**a**) Point cloud filling example of the shoulder’s BS. Red: original points; green: artificial points created to fill vertical gaps around pointy areas; and cyan: artificial points created to fill the bottom part of the BS. (**b**) Alpha shape computed on the shoulder’s BS. (**c**) Monte Carlo points of the shoulder’s BS belonging to its alpha shape.

**Figure 8 sensors-25-01515-f008:**
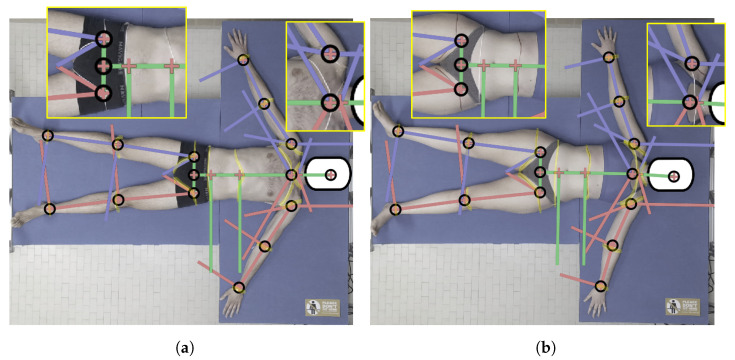
Images of the subjects with section lines that were directly drawn by an expert on the skin using a pen for the female subject and a tape for the male subject (which is highlighted in the picture). Vectors computed from the KPs were superimposed on the image to highlight any discrepancy. (**a**) Male subject, (**b**) female subject.

**Figure 9 sensors-25-01515-f009:**
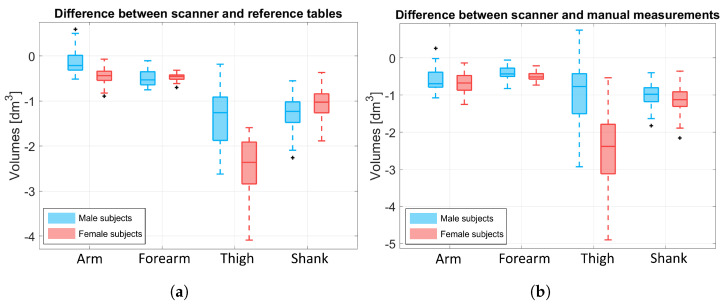
Boxplot of the resulting ΔV values. Comparison was only made for the limbs. Blue boxes refer to male data and pink boxes to female data. (**a**) Difference between the scanner volume and the one obtained from the literature reference, $\Delta V_{scanner-ref,BS}$. (**b**) Difference between the scanner volume and the one obtained by approximating the BS to a truncated cone using the subject’s manual measurements, $\Delta V_{scanner-cone,BS}$.

**Figure 10 sensors-25-01515-f010:**
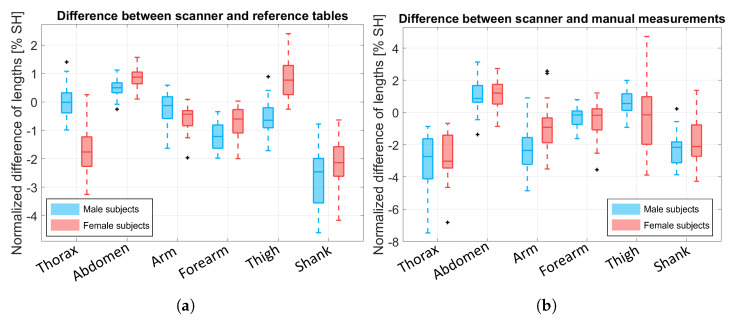
Boxplot of the resulting ΔL values, which were normalized over the subject’s height (SH). Blue boxes refer to male data and pink boxes to female data. Outliers are depicted in black with a plus symbol. (**a**) Difference between the BS lengths obtained from our biomechanical model and the BS lengths obtained from the literature reference, $\Delta L_{scanner-ref,BS}$. (**b**) Difference between the BS lengths obtained from our biomechanical model and the BS lengths obtained from manual measurements, $\Delta L_{scaner-cone,BS}$.

**Figure 11 sensors-25-01515-f011:**
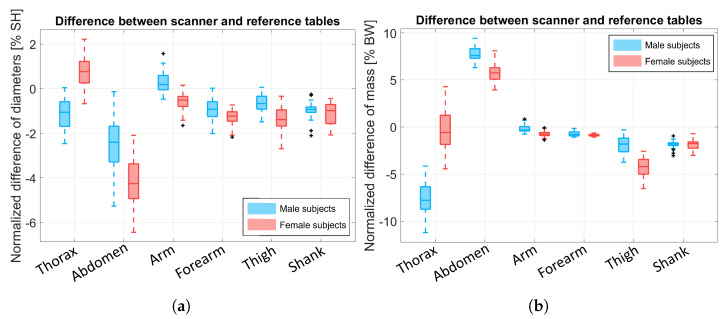
(**a**) Boxplot of the $\Delta D_{BS}$ computed as the difference between ${\hat{D}}_{scan,BS}$ and ${\hat{D}}_{ref,BS}$. The difference was normalized over the subjects’ height (SH). (**b**) Difference of the BSs mass resulting from our scanner-based method and the reference tables. The data were normalized over the subject’s body weight (BW).

**Figure 12 sensors-25-01515-f012:**
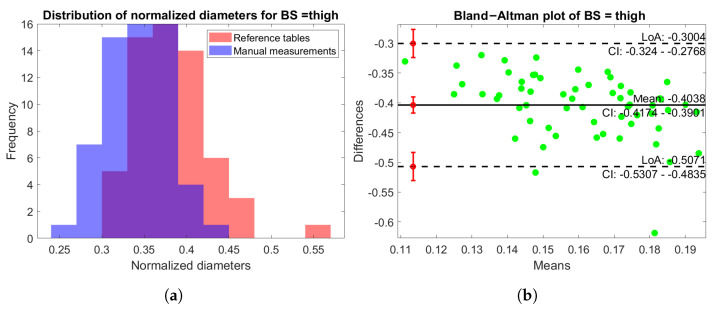
(**a**) Histogram distribution of the normalized diameters for both the rescaled table values (red) and manual measurements (blue) for the leg BS. (**b**) Bland–Altman plot of the normalized diameters of the thigh BS.

**Table 1 sensors-25-01515-t001:** Summary of the manual measurements taken from the subjects during the experiment. All measurements were taken with the subjects standing still and undressed (except for underwear) from the front of the body except when stated otherwise.

Body Segment	Length Measurements	Width Measurements	Circumferences
Head	Length $L_{head}$ taken from C7 to the head tip. This measure was taken from the back of the body.	//	Circumference $O_{head}$ taken 1 cm higher than the ears.
Arm, forearm, leg, shank	Segment’s length, $D_{segment}$, taken in between the proximal and distal joints.	//	Circumferences taken in correspondence with the proximal and distal joints $O_{p}$ and $O_{d}$.
Trunk	Length of the trunk, $L_{trunk}$, taken from C7 to T10. This measure was taken from the back of the body.	Distance between the shoulders, $D_{shoulders}$, taken in correspondence with the shoulder joints.	Circumference of the chest, $O_{chest}$, taken in correspondence of the nipples.
		Width of the sternum taken considering the sternum’s sides, $W_{sternum}$.	Circumference of the sternum, $O_{sternum}$.
Abdomen	Length of the abdomen, $L_{abdomen}$, taken from T10 to L5. This measure was taken from the back of the body.	Width in correspondence with T10, $W_{T10}$, and taken by considering the sides of the body.	Circumference of the abdomen, $O_{T10}$, taken in correspondence with T10.
Pelvis	//	Distance between the two asi, $D_{asi}$, taken in correspondence with the asis’ joints.	Circumference of the hips taken in correspondence with the asis, $O_{asi}$.
		Width of the hips taken considering the sides of the body, corresponding to the asis position.	
		Width of the trochanters, $W_{troch}$, taken from the frontal position of the trochanter to the back. This is considered a “depth” measure.	
		Vertical distance from the asi to the trochanter, $D_{asi-troch}$.	
Hand	Length of the right hand, $L_{hand}$, taken from the medium finger tip to the wrist joint.	//	//
	Width of the right hand, $W_{hand}$, taken considering when the fingers were close together, from the thumb joint to the other side.		
Foot	Length of the right foot, $L_{foot}$, taken from the heel to the toes.	//	Circumference of the ankle, $O_{ankle}$.
	Width of the right foot, $W_{foot}$, taken from the big toe joint to the other side (maximum width).		
	Height of the heel taken from the ankle to the ground, $H_{foot}$.		

**Table 2 sensors-25-01515-t002:** The averages and standard deviations of the subjects’ age, height, mass, and BMI, divided by gender.

	Females		Males	
	**Mean**	**Std**	**Mean**	**Std**
Age [years]	28.5	6.5	26.6	4.1
Mass [kg]	61.1	7.4	74.1	10.7
Height [cm]	166.3	7.2	176.4	6.5
BMI [kg/m^2^]	22.1	2.8	23.7	2.5

## Data Availability

Data are available on request.

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
