# Peer review of "Measurement of Human Body Segment Properties Using Low-Cost RGB-D Cameras"

_sensors, 2025, doi:10.3390/s25051515_

Round 1

Reviewer 1 Report

Comments and Suggestions for Authors

While it is not a challenging task to automatically and accurately estimate human body volume and mass using precise 3D body scan models under current technological conditions, the proposed method in this paper, which utilizes low-cost RGB-D cameras, lacks novelty and accuracy. The specific shortcomings are as follows:

1. RGB-D cameras are not accurate 3D scanning devices. The quality of their point clouds is even lower than that of common low-cost MVS (Multi-View Stereo) systems. Therefore, if this paper is to be published, a comparative evaluation of point cloud quality should be provided.

2. The human subject is lying on a platform that is not transparent. Therefore, the part of the human body in contact with the platform is not scanned, and this data loss will lead to systematic errors. The authors did not provide a reasonable correction or compensation for this.

3. The 3D point cloud extracted from a single depth map of a single RGB-D camera is the worst type of point cloud, and cannot even compare with fusion models. Conducting subsequent research on such low-quality and noisy point clouds, the errors in the original signal are difficult to correct.

4. The skeleton estimation in this paper does not correspond to the actual human skeleton. The authors also failed to propose a method for automatically finding the real human skeleton/joints, but only performs linear reasoning and calculation on the extracted skeleton. The rationality of this approach lacks a large amount of empirical verification.

5. Based on accurate 3D scanning, the volume of the human mesh can be accurately calculated through numerical integration. Compared with this, the method proposed in this paper is not advanced.

In summary, this paper is more like an experimental report and lacks novelty and accuracy. It is not recommended for publication.

Author Response

Comment 1: RGB-D cameras are not accurate 3D scanning devices. The quality of their point clouds is even lower than that of common low-cost MVS (Multi-View Stereo) systems. Therefore, if this paper is to be published, a comparative evaluation of point cloud quality should be provided.

Response 1: Although we may generally agree with the statement about RGB-D cameras being less accurate compared to other scanning technologies (e.g., laser scanning or structured light), we do not agree with the adoption of MVS systems for measurement applications. The reason stands in the operating principle of the two approaches: MVS adopt multiple 2D views and infer by calculation the original 3D shape of the object, while in the case of Kinect Azure (which we used) the measuring principle is time-of-flight, hence a direct measure of the original 3D shape through the timing of the pulsed IR light that hits the object. From a measurement point of view, this approach is more suitable since it’s a direct measure. Moreover, the adoption of MVS in such a context is not viable due to limitations of space and time. The set-up was mounted in a hospital room and ideally the subjects would not lay still on the bed for more than a minute, since the target subjects are people with motion impairments or neurological conditions affecting the body structure and movement, thus explaining the necessity of a bed instead of doing the scan while standing, thus obtaining also the back part. To emphasize the rationale behind the design choices of this work, we expanded the Introduction section (lines 59-78). Furthermore, when dealing with accuracy it is necessary to specify what level of accuracy (and resolution) is needed for the specific application. We do agree that uncertainty in the measurment of a mechanical small object for precision manufacturing should be the lowest possible, reaching even micrometers for some special objects, but in the case of biomechanics (where the main goal is to obtain the human body segments’ kinematics), lower accuracies in the volume measure of the body segments are acceptable if kept in an acceptable range. Kinect Azure is accurate enough for this purpose. Finally, we want to stress that an evaluation of the Kinect Azure measurement robustness and accuracy was conducted in comparison with the previous Kinect v2, highlighting the improved performance of the device especially in indoor environments. We encourage the reviewer to see ref. 27 and other references cited in that article for more information about Kinect Azure performance.

Comment 2: The human subject is lying on a platform that is not transparent. Therefore, the part of the human body in contact with the platform is not scanned, and this data loss will lead to systematic errors. The authors did not provide a reasonable correction or compensation for this.

Response 2: Thank you for this comment. As mentioned in the introductory paragraph of the paper, this work aimed at the development of a measurement set-up to estimate body segments volume and mass of the scanned subject. The specific application for which it was developed is relative to patients with motion impairments or neurological conditions that affect their bodies and movement abilities, thus it is impossible for them to stand and, for some, it is also quite difficult to stay still in position for more than a minute. As a result, we needed a fast scanning solution and a comfortable design of the set-up, thus explaining the choice of Kinect Azure and of the horizontal bed. This motivation has been stressed again in the revised version of the paper at lines 59-78. In addition, we want to stress that, even if the bed was transparent, the back of subjects would be deformed anyway since they were lying on the bed instead of standing, so setting a second camera to register the back would lead to useless data. The proposed solution is obviously a compromise (we do lack the back portion of the body, after all), but is a working one that produced contained errors in the estimation. We will work in the future to improve the set-up and apply it to affected patients. This work was conducted on healthy subjects to first validate the measurement pipeline before conducting a more thorough research on patients, since it is quite difficult to recruit them and obtain the required paperworks for approval from the ethical committee.

Comment 3: The 3D point cloud extracted from a single depth map of a single RGB-D camera is the worst type of point cloud, and cannot even compare with fusion models. Conducting subsequent research on such low-quality and noisy point clouds, the errors in the original signal are difficult to correct.

Response 3: We understand the reviewer’s concern. However, it depends on the magnitude of errors and noise for the specific application. Kinect Azure is actually quite robust to them if used indoor and after conducting the appropriate warm-up and calibration steps (as highlighted by ref. 27). Moreover, the procedure conducted to extact the point cloud was compliant with the source code of the camera SDK, ensuring the lowest discrepancies and noise possible. Redundancy of point clouds obtained from several devices was tested as well with no relevant improvements in the overall accuracy of volume and mass of the body segments; hence, to keep everything simple and quick, we decided to keep only one device. This outcome was actually expected since the main drawback of the set-up is that the subject is lying on a bed, thus the back portion is not visible and points belonging to the lateral portions of the body (e.g. the sides of the torso) are scarce. However, as explained before, this was a necessary choice and it was partially corrected by software as detailed in the paper (Section 2).

Comment 4: The skeleton estimation in this paper does not correspond to the actual human skeleton. The authors also failed to propose a method for automatically finding the real human skeleton/joints, but only performs linear reasoning and calculation on the extracted skeleton. The rationality of this approach lacks a large amount of empirical verification.

Response 4: As stated in Section 2.5, we used a combination of a skeletonization model (Blazepose) that is not based on the actual human structure of the body, corrected afterward according to the biomechanical model definition detailed in the EUROBENCH project (ref. 25). This model is not empirical, because it is derived from the anthropometric data of literature medical research (ref. 10 and 32), reporting the average size of the vertebrae. Hence, the proposed segmentation (see Fig. 2b) is directly obtained from a combination of the two approaches. The topic of estimating the actual human skeleton from 2D images deserves a detailed publication separate from this one, and it was not the focus of this work. The proposed corrections derived from the skeletonization model applied (Section 2.4 and 2.5) are sufficient for our objective and, as demonstrated by the experiment detailed in Section 3.1, the estimation is the same as the one made by an expert while the subject was lying on the bed (represented by markers applied manually on the body). If the objective of this work was the definition of a new skeletonization model to infer the actual skeleton joints position, different and more exhaustive experiments would have been conducted.

Comment 5: Based on accurate 3D scanning, the volume of the human mesh can be accurately calculated through numerical integration. Compared with this, the method proposed in this paper is not advanced.

Response 5: The accurate 3D scanning mentioned by the reviewer refers to photogrammetry or MVS systems, either way dense point clouds that allow the estimation of accurate meshes. The proposed approach deals with point clouds with a low number of points, hence they are not comparable. In addition, the objective is not to define a method to compute the volume of an object from its point cloud in a generic way, but instead the computation of the volume of a human body segment, which is a portion of a bigger point cloud.

Comment 6: In summary, this paper is more like an experimental report and lacks novelty and accuracy. It is not recommended for publication.

Response 6: We are sorry to read this. However, we do not agree with the reviewer, since no other works (to our knowledge) dealt with the topic of obtaining human body segments volumes and mass for the targeted application (the measurement of patients with motion impairments) and conducted a thorough comparison of results with existing methods adopted by healthcare professionals (e.g., traditional antrophometric tables that healthcare professionals still use today and cone segments). Furthermore, the necessary accuracy for such application is not extensive and it is easily reached by the Kinect Azure device adopted, thus, we do not understand the concern of the reviewer about accuracy, especially since no values were provided to understand the magnitude of accuracy requested (eg. Nanometers, micrometers, millimeters?) compared to what was provided in the paper, and on what estimation (volume or segments’ length?). It is quite obvious that more expensive devices and approaches, such as tomography or photogrammetry, can reach more detailed results; however, the objective was to keep everything simple and affordable and we think we have succeeded.

Reviewer 2 Report

Comments and Suggestions for Authors

Overall, this is an interesting piece of work, which attempts to address a difficult issue present within the field of 3D imaging and body measurement, that of collecting data from people who cannot stand upright/unsupported and I commend the authors for their efforts in this regard.

I have a few comments on the methodology, which I detail below:

- In the methods section you describe how manual measures of the participants were collected first, before then collecting the 3D data. However, it was not specified how the manual measures were collected. Most importantly, it was not specified whether these measures were collected while the participants were standing or lying down. Please provide further details for this section of the experimental protocol.

- Further, if the manual measures were collected whilst the participant were in any position that differs from the position they were in whilst the 3D measures were collected (lying down) this would have a significant effect on the comparability between measures collected manually and via 3D imaging (circumferences, volumes, lengths, etc.). Please provide further details about the positioning of participants in all stages of measurement and the efforts employed to ensure their comparability between stages.

- The method for collecting 3D data from participants employed in this study (whilst the participants lays flat on a bed) will inevitably lead to issues, which the author does discuss briefly in the manuscript. The main issues being the fact that the act of lying flat causes flattening of the posterior aspect of the body, which will change the eternal shape features of the body and any measures derived from the 3D data (circumferences, volumes, etc.). In addition, laying flat on your back will cause areas of the anterior aspect of the body to also flatten, particularly in female participants as well as males with greater amounts of adipose tissue, compounding the error in measurement. Also, positioning the participant in this way completely occludes the posterior aspect of the participant, which leads the author to have to resort to methods to reconstruct the posterior aspect of their body, which has several limitations. Therefore, more detail needs to be provided in the introduction section about the issues that participant posture and positioning has on collecting measurements, both manually and using 3D methods. Also, the author should propose additional alternative methods for positioning participants during 3D measurement that solves some of the issues raised here - e.g. can 3D data be collected whilst the participant lays on their front and their back with data collected from both positions being combined together to give a full 3D image? Can the participant be supported to be in a vertical posture using postural aids, or reclining beds that elevate them from laying in a prone position? 

Author Response

Comment 1: In the methods section you describe how manual measures of the participants were collected first, before then collecting the 3D data. However, it was not specified how the manual measures were collected. Most importantly, it was not specified whether these measures were collected while the participants were standing or lying down. Please provide further details for this section of the experimental protocol.

Response 1: We thank the reviewer for this comment. The manual measurements taken were described in publication 26, which was our previous work. To avoid confusion and improve clarity, we decided to add the corresponding information requested in section 2.1 (greatly modified) and Table 1, explaining how each manual measure was obtained.

Comment 2: Further, if the manual measures were collected whilst the participant were in any position that differs from the position they were in whilst the 3D measures were collected (lying down) this would have a significant effect on the comparability between measures collected manually and via 3D imaging (circumferences, volumes, lengths, etc.). Please provide further details about the positioning of participants in all stages of measurement and the efforts employed to ensure their comparability between stages.

Response 2: We understand the reviewer's concern. Manual measurements were taken with the subjects standing still, as it is the standard procedure followed by healthcare professionals (also explained in more detail in the modified version of Section 2.1, lines 102-119). We do agree that comparing a manual measurement taken standing still with the one taken laying down is notably different. However, this is how healthcare professionals operate, especially on healthy patients, and it is considered the standard in the field. In addition, literature research highlights how there is minimal difference between manual measurements taken standing still versus lying horizontally, especially if the adipose tissue of the subject is contained (see references 28, 29 and 30). In addition, a key objective was to compare the two approaches (standard versus scanner). The differences mentioned by the reviewer are in fact a source of error that affect our results, especially for those body parts that are less compact and muscular, like the torso and abdomen. This issue is now stressed in Section 3.3 (see lines 403-409). 

Comment 3: The method for collecting 3D data from participants employed in this study (whilst the participants lays flat on a bed) will inevitably lead to issues, which the author does discuss briefly in the manuscript. The main issues being the fact that the act of lying flat causes flattening of the posterior aspect of the body, which will change the eternal shape features of the body and any measures derived from the 3D data (circumferences, volumes, etc.). In addition, laying flat on your back will cause areas of the anterior aspect of the body to also flatten, particularly in female participants as well as males with greater amounts of adipose tissue, compounding the error in measurement. Also, positioning the participant in this way completely occludes the posterior aspect of the participant, which leads the author to have to resort to methods to reconstruct the posterior aspect of their body, which has several limitations. Therefore, more detail needs to be provided in the introduction section about the issues that participant posture and positioning has on collecting measurements, both manually and using 3D methods. Also, the author should propose additional alternative methods for positioning participants during 3D measurement that solves some of the issues raised here - e.g. can 3D data be collected whilst the participant lays on their front and their back with data collected from both positions being combined together to give a full 3D image? Can the participant be supported to be in a vertical posture using postural aids, or reclining beds that elevate them from laying in a prone position? 

Response 3: We thank the reviewer for the interesting suggestions proposed to improve our work. To address the mentioned issues relative to the set-up (which was necessary since the target application involves patients with motion impairments), we expanded the introduction and conclusion section, also proposing some workarounds to improve the point cloud collection procedure in the future (lines 59-78, 102-119, 142-145, 403-409, 493-496). However, the proposed solutions may not be applicable depending on the patient condition, so to make stronger claims about the adoption of those workarounds we would need to run an experimental campaign with patients and test them, taking into account their comfort. This could be a topic for the future.

Reviewer 3 Report

Comments and Suggestions for Authors

1. The author must indicate in which application this method would be useful and the errors and acceptable. Given that the experiment was done in a very controlled, error with this method in a non controlled environment and including clothing will seriously affect the results.

2. Line 307: "The section lines were drawn on the skin using either a pen or a tape". The measurement was taken one 1 male and 1 female, please write the exact procedure to avoid ambiguity. please check for other sentences (example figure 8) to avoid ambiguity.

3. RGB-D camera resolution for RGB and D should be included.

4. Given that the resolution is relatively low when using it to measure whole human body and given such a controlled experimental condition, markers could have been used to increase accuracy.

5. The simple stats of the participants such as age, stature and anthropometric measurements must be included to understand the results.

6. It is not clear how the manual volume measurements (as shown in Figure 9b) of the body parts obtained in dm3. 

7. Instead of using Blue and pink in the figure label, a legend in the plot might be easier to visualize. 

8. In Figure 9 the Y-axis unit seems to be relative values rather than absolute values.

9. In figure 10-11, what is SH?

10. In figure 13 what is BW?

11. It would be good to have measurement values in addition to % relative values. As % values does not show the actual error in terms on units such as mm or Kg. The accuracy will influence the application of the proposed method. 

Author Response

Comment 1: The author must indicate in which application this method would be useful and the errors and acceptable. Given that the experiment was done in a very controlled, error with this method in a non controlled environment and including clothing will seriously affect the results.

Response 1: The target application of our measurement set-up involves patients with motion impairments or other conditions that affect their body and movement ability. This was discussed in the introduction paragraph, now expanded to make it clearer, and stressed several times in the whole document (lines 59-78, 102-112, 142-145, 403-409). In the results section we already discussed the errors obtained which are mostly due to the subjects lying on the bed, thus the back portion of their body is not scanned, and not on the accuracy of the scanning device. However, given the intended applications, subjects have to lay on the bed and so those errors should be addressed afterwards by software or by proposing a different measurement procedure, as discussed in the modified version of the conclusions paragraph (lines 403-409). Finally, the intended use of the set-up does not involve clothes except for the necessary underwear, so the issue of clothes is not relevant in this case (as stated in section 2.1 detailing the protocol). 

Comment 2: Line 307: "The section lines were drawn on the skin using either a pen or a tape". The measurement was taken one 1 male and 1 female, please write the exact procedure to avoid ambiguity. please check for other sentences (example figure 8) to avoid ambiguity.

Response 2: Thank you for this correction. We edited the text as suggested. 

Comment 3: RGB-D camera resolution for RGB and D should be included.

Response 3: Thank you for this comment. We edited the text as suggested at lines 147-153.

Comment 4: Given that the resolution is relatively low when using it to measure whole human body and given such a controlled experimental condition, markers could have been used to increase accuracy.

Response 4: Thank you for this suggestion. In the early stages of this project, markers were considered as well for this purpose but were excluded because they do not significantly improve the accuracy of the point cloud acquisition while contaminating the body in return. They could be helpful in the case of multi-scans that need to be aligned afterwards, but in our case we only get a single scan of the subject. As a result, the adoption of markers is briefly discussed in the conclusions paragraph (lines 403-409) as a mean to, in future developments, improve the measurement in the case of workarounds required to acquire the back portion of subjects with motion impairments.

Comment 5: The simple stats of the participants such as age, stature and anthropometric measurements must be included to understand the results.

Response 5: We added the average age, weight, height and BMI of participants divided by gender in Table 2, as well as the corresponding standard deviations.

Comment 6: It is not clear how the manual volume measurements (as shown in Figure 9b) of the body parts obtained in dm3.

Response 6: The procedure is explained in Section 3.2, specifically equations 8 and 9. The volume obtained from antrophometric tables is straightforward to compute given that ref 10 contains the data needed (mass and density of the body segment). For the volume derived from manual measurements we applied a standard procedure adopted by professionals of the field, based on the computation of a truncated cone section as explained in Section 3.2 and highlighted by equation 9. The terms involved are the length of the body segment (in dm) multiplied for squared radii terms (in dm2), thus obtaining dm3. The formulas for the computation of truncated cones are common geometric knowledge. In addition, information about how the manual measurements were taken are now included in Table 1.

Comment 7: Instead of using Blue and pink in the figure label, a legend in the plot might be easier to visualize. 

Response 7: The legend was added in the plots as requested. 

Comment 8: In Figure 9 the Y-axis unit seems to be relative values rather than absolute values.

Response 8: Thank you for noting this. Yes, the graph shows delta values, e.g., differences between two quantities, specifically the volume obtained from our scanner-based procedure and the reference/truncated cone volumes. This was abundatly explained in both the text and the caption; however, to improve clarity, we modified Section 3.2 adding a few lines about the computation of the delta quantities and their formulas (eq. 6-7). The same was done in the following sections dealing with similar quantities. Please keep in mind that absolute differences show greater variability if the original data is not normalized over the subject’s height or weight accordingly.

Comment 9: In figure 10-11, what is SH?

Response 9: It refers to Subject’s Height, as mentioned in the text. We added it in the caption for clarity. 

Comment 10: In figure 13 what is BW?

Response 10: It refers to Body Weight, as mentioned in the text. We added it in the caption for clarity.

Comment 11: It would be good to have measurement values in addition to % relative values. As % values does not show the actual error in terms on units such as mm or Kg. The accuracy will influence the application of the proposed method. 

Response 11: Thank you for this comment. For lengths and equivalent diameters the original values of each subject were normalized according to the corresponding subject’s height, while for the masses the same was done by normalizing over the subject’s weight. Normalization is fundamental since it allows comparison between values (e.g. an error of 3 kg in the mass estimation of a very thin and short person is quite different compared to the same 3 kg error for a tall and muscular person). Moreover, we showed our results in the form of boxplots divided by gender and body segment instead of showing the results of each subject individually. In this case we could have reported both the % error and the corresponding absolute error, but since we have a population of subjects we cannot correlate the % error shown in the boxplot to the corresponding absolute value because it’s subject-specific. A workaround is to multiply the % error for the average weight or height of the population, but this will result in a rough approximation that we prefer not to include in the article. Moreover, a % error says more than the absolute value in our opinion, since variables such as weight and height are removed and do not affect the quantity. Healthcare professionals can easily understand if a 4% error on the mass estimation is too much by multiplying it for the subject’s weight.
To better explain the necessity of normalization, we added a few lines in the results section (lines 392-396).

Round 2

Reviewer 1 Report

Comments and Suggestions for Authors

stimating the volume of different human body parts is valuable, and the research topic is interesting. However, this paper cannot be accepted for publication due to the following main reasons:

The literature review is insufficient. Only three papers published after 2022 were cited, which does not adequately reflect the state of the art. Recent studies on volume estimation using RGB-D sensors are missing. For example, it is easy to find on Google Scholar that Hu et al. introduced a deep learning approach to estimate the volume of each body segment [A], addressing the same problem as this study.

[A] Hu, P., Dai, X., Zhao, R., Wang, H., Ma, Y. and Munteanu, A., 2023. Point2PartVolume: Human body volume estimation from a single depth image. IEEE Transactions on Instrumentation and Measurement, 72, pp.1-12.

This manuscript is an extension of Reference [26]. However, the primary distinction from Reference [26] is that the authors have utilized the Kinect Azure sensor. It seems that no novel algorithms have been developed in this study. Moreover, the method for estimating volume is not sufficiently justified, as it relies on a single-view scan without incorporating shape completion techniques.

Author Response

Q1: The literature review is insufficient. Only three papers published after 2022 were cited, which does not adequately reflect the state of the art. Recent studies on volume estimation using RGB-D sensors are missing. For example, it is easy to find on Google Scholar that Hu et al. introduced a deep learning approach to estimate the volume of each body segment [A], addressing the same problem as this study.

[A] Hu, P., Dai, X., Zhao, R., Wang, H., Ma, Y. and Munteanu, A., 2023. Point2PartVolume: Human body volume estimation from a single depth image. IEEE Transactions on Instrumentation and Measurement, 72, pp.1-12.

A1: We thank the reviewer for noticing this issue. We reworked the Introduction section to incorporate recent works on the topic and briefly discuss their contents in comparison with the proposed one. Edited lines are: 19-21, 32-36, 38-43, 62-78, 107-130 (highlighted in yellow in the manuscript with highlights provided as supplementary material). We purposely incorporated works that exploit 3D data to obtain an estimation of the final volume, either of the whole body or of body parts, since 2D-based approaches are out of scope with respect to the proposed method.

Q2: This manuscript is an extension of Reference [26]. However, the primary distinction from Reference [26] is that the authors have utilized the Kinect Azure sensor. It seems that no novel algorithms have been developed in this study.

A2: We understand the reviewer’s concern. As a consequence, we edited the introduction section to stress more evidently the advantages of the proposed work in contrast with the previous conference presentation and with existing literature. The edited lines are 107-130. Despite proposing a methodology similar to the previous work, the main extensions are (i) the different device, (ii) the higher number of tested subjects (60 instead of 30), (iii) the validation procedure conducted which is based on a novel metric named "equivalent diameters". The validation section is actually the core of the work and it's the novel part often missing from other works on this matter.

Q3: Moreover, the method for estimating volume is not sufficiently justified, as it relies on a single-view scan without incorporating shape completion techniques.

A3: We agree with the reviewer that the obtained point cloud lacks the back points since the subjects are lying on a bed instead of standing still. This is the major drawback of the presented work, as we already discussed in several sections of the article. Even by adopting multiple cameras to obtain other views of the body (an experiment we tried without success) the resulting point cloud would still be lacking the back points due to the measurement set-up configuration. However, this is a feature that cannot be changed considering that for the EUROBENCH project (funding this research) the target subjects are people with motion impairments that cannot stand still and may have delicate bodies requiring comfortable bedding during acquisition. This issue was stressed several times in the article as well. As for the shape completion techniques, the proposed method is based on point cloud filling (Section 2.9). This is evidently an approximation and the method would definitely benefit from advanced shape completion techniques that could recreate the missing portion of the body back. This problem was discussed in the results and conclusions sections as well and potential solutions were provided to overcome it in future works. Despite this drawback, however, the proposed method is fast and sufficiently accurate for the limbs to justify a publication, especially since the main focus of the work lays on its validation. In fact, compared with other works on the matter, we proposed the segmentation of the point cloud using a biomechanical model into 16 body segments (thorax and chest separated at first but joined for the volume and mass computation). Volume computation was conducted using the Monte Carlo method to count the number of points inside the alpha shape of the body segment. Finally, the experiments were conducted on a gender-balanced dataset of real people scanned for this purpose (not synthetic models nor point clouds from datasets) and the system’s performance was evaluated in comparison with antrophometric data and manual measurements proposing a novel metric named “equivalent diameter”. The evaluation section of the paper is the actual core of the work, often overlooked in several other methods. In particular, deep learning-based approaches that do reconstruct missing parts of the original single-view scan work only with standing subjects and, since they’re trained on healthy subjects, could potentially lead to incorrect estimations of the missing portions of the body for diseased patients. Future contributions will investigate the matter further; however, we still think that the proposed work is an advancement in the field (however small) and thus deserving a publication.

Reviewer 3 Report

Comments and Suggestions for Authors

"The proposed work focuses on people who cannot stand; thus, they cannot be subjected to time-consuming acquisition procedures or measurement methods that are as sophisticated as they are expensive and therefore not widely used."

The subject used are healthy subjects : "A total of 60 healthy subjects (30 males, 30 females) participated in the study" ... "Exclusion criteria were inability to lay down and stay still on the bed"

If the proposed study was for people who cannot stand, then need to have strong justification why this type of subject was not used and healthy subjects were used. Otherwise, the study will lack external validity. 

Also it is better to add the need for these type of research for the given population. 

Author Response

Question 1: If the proposed study was for people who cannot stand, then need to have strong justification why this type of subject was not used and healthy subjects were used. Otherwise, the study will lack external validity.

Response 1: As the reviewer noted, this work was designed for patients that cannot stand. First, from a measurement point of view, it is fundamental to first validate a measurement pipeline on measurands that can be refrenced to a ground-truth (in our case, the anthropometric tables that were based on healthy subjects and the truncated cones model). If the measurement system to be tested was not even capable of obtaining the required outcomes (volumes and masses of body segments) on healthy patients, the research on ill subjects would have been impossible.

Moreover, patients with motion impairments are not readily available for experimentation, and institutions require strict ethical approvals to recruit patients for research purposes. These are the main reasons why this work involves healthy subjects. Working with patients with motion impairments is our final goal that will be addressed in the future, for which the results on healthy subjects will be considered as a reference. To make a stronger claim that justifies the approach, we added the following in the introduction paragraph:

"However, before conducting extensive experiments on patients, it is of utmost importance to first validate the measurement pipeline proposed on healthy subjects. This is the reason why the present work describes the validation of the developed measurement pipeline on healthy subjects, focusing on the comparison with existing methods used by healthcare professionals and by the biomechanics research community."

And in the conclusions paragraph we added the following:

"The proposed measurement system is specifically designed to work with subjects with motion impairments. The present work describes the experimental campaign and results conducted to validate the system on healthy subjects, a fundamental requirement before conducting extensive research on the target patients."

[...]

"Results highlight that, despite the necessary improvements required to optimize the volume and mass estimation for the trunk, abdomen, and pelvis BSs, the developed system is a valid alternative to traditional practices such as anthropometric tables and the truncated cones model, even for healthy subjects. As a result, further experiments will involve patients with motion impairments to demonstrate the validity of the developed system on the target subjects. In the case of such subjects, obtaining a full body scan may be difficult or even impossible according to their physical conditions; hence, a workaround to obtain it may be to design dedicated equipment to support them in an upright or partially standing position, coupled with the adoption of markers to facilitate the front and back point cloud fusion."

Please check the manuscript with highlights in yellow to quickly visualize the edits made at this round of revisions.

Question 2: Also it is better to add the need for these type of research for the given population. 

Response 2: This research work will benefit the biomechanics and healthcare research community that work with rehabilitation of patients with motion impairments and on the study of movement kinematics (e.g. gait analysis), as well as engineers working on the design of effective exoskletons that aid the patient during walking. This was extensively explained in the introduction and in the conclusion paragraphs.

Both kind of patients (healthy and ill) benefit from this research, since the developed system is a fast and cost-effective solution to obtain volume and mass of their body segments. Moreover, our research highlighted how gender-specific biomechanical models need to be designed to address specific differences between subjects of different genders, a topic often overlooked in this kind of research. All of this information was already written in the introduction and conclusion paragraph as well.